# Dendrimers: Advancements and Potential Applications in Cancer Diagnosis and Treatment—An Overview

**DOI:** 10.3390/pharmaceutics15051406

**Published:** 2023-05-04

**Authors:** Andreea Crintea, Alexandru Cătălin Motofelea, Alina Simona Șovrea, Anne-Marie Constantin, Carmen-Bianca Crivii, Rahela Carpa, Alina Gabriela Duțu

**Affiliations:** 1Department of Molecular Sciences, Faculty of Medicine, Iuliu Hațieganu University of Medicine and Pharmacy, 400349 Cluj-Napoca, Romania; 2Department of Internal Medicine, Faculty of Medicine, Victor Babeș University of Medicine and Pharmacy, 300041 Timișoara, Romania; 3Department of Morphological Sciences, Faculty of Medicine, Iuliu Hațieganu University of Medicine and Pharmacy, 400000 Cluj-Napoca, Romania; 4Department of Molecular Biology and Biotechnology, Faculty of Biology and Geology, Institute for Research-Development-Innovation in Applied Natural Sciences, Babeș-Bolyai University, 400084 Cluj-Napoca, Romania

**Keywords:** cancer, dendrimers, diagnosis, treatment, targeting, photothermal therapy, photodynamic therapy, gene transfection

## Abstract

Cancer is a leading cause of death worldwide, and the main treatment methods for this condition are surgery, chemotherapy, and radiotherapy. These treatment methods are invasive and can cause severe adverse reactions among organisms, so nanomaterials are increasingly used as structures for anticancer therapies. Dendrimers are a type of nanomaterial with unique properties, and their production can be controlled to obtain compounds with the desired characteristics. These polymeric molecules are used in cancer diagnosis and treatment through the targeted distribution of some pharmacological substances. Dendrimers have the ability to fulfill several objectives in anticancer therapy simultaneously, such as targeting tumor cells so that healthy tissue is not affected, controlling the release of anticancer agents in the tumor microenvironment, and combining anticancer strategies based on the administration of anticancer molecules to potentiate their effect through photothermal therapy or photodynamic therapy. The purpose of this review is to summarize and highlight the possible uses of dendrimers regarding the diagnosis and treatment of oncological conditions.

## 1. Introduction

### 1.1. Generalities

Dendrimers are nanomaterials with unique properties used in cancer diagnosis and treatment, targeting tumor cells, controlling the release of anticancer agents, and combining anticancer strategies [1]. Cancer is a malignant condition characterized by the uncontrolled proliferation of atypical cells. Cancer progression is supported by the imbalance or damage of proto-oncogenes that encode proteins involved in the development and differentiation of tumor cells, but also tumor suppressor genes that encode proteins that produce inhibitory signals for cells in need and produce apoptosis [2]. According to the World Health Organization, cancer is the leading cause of death globally, and lung cancer is the most common cause of cancer-related death [3,4]. As the mortality rate associated with cancer is increasing, research on cancer therapies is growing, and identifying the most effective treatment method is the goal of most scientists [5,6]. The main treatment methods for this group of pathologies include surgical excision, chemotherapy, radiotherapy, and immunotherapy. The effectiveness of these therapies depends on the severity of the disease and the particular reaction of each organism. However, a major problem encountered in patients undergoing the above-mentioned therapies is that these therapies cannot distinguish between tumor cells and healthy cells in the body, resulting in the occurrence of serious adverse reactions [5,7,8,9]. These adverse reactions are most often represented by alopecia, decreased immunity through the suppression of bone marrow function, fever, nausea, vomiting, hepatotoxicity, cardiotoxicity, neurotoxicity, electrolyte imbalances, and decreased muscle tone [10,11]. One way to avoid these adverse reactions is to use targeted therapies, where drugs or mechanisms for destroying tumor cells are specifically directed to the tumor [8,12,13]. Targeted therapy is a new-generation chemotherapy that seeks to target certain proteins or genes specifically linked to a particular cancer or tumor vasculature that promotes its growth [2]. Nanotechnology is an option for such therapies and is based on the synthesis of “tools” that act as transporters of drugs to a specific tumor target [14,15,16,17]. This review aims to give an overview of the potential diagnostic and therapeutic applications of dendrimers for oncological conditions.

### 1.2. Methods

The process of research for this review involved a thorough and precise search for relevant bibliographic sources across several databases, such as PubMed, Google Scholar, Cochrane, and Embase. The objective was to identify potential sources that contain information on the utilization of nanoparticles and dendrimers in cancer diagnosis and treatment. To conduct the search, the authors used a combination of appropriate keywords, including nanoparticles, dendrimers, cancer diagnosis, cancer treatment, and targeting. The search process was comprehensive, and the authors considered all relevant sources that appeared during the search. Initially, the authors identified over 227 bibliographic sources, which they evaluated carefully to determine their relevance to the study’s focus. Only sources whose information was pertinent to the research question and whose results were consistent with similar studies were included. As a result, the authors narrowed down the pool to 217 sources, which were considered relevant and included in the final review. The methodology used by the authors was designed to ensure a rigorous and comprehensive analysis of the available literature on the use of dendrimers in cancer diagnosis and treatment. The inclusion criteria were strict to ensure that only high-quality and relevant sources were included in the review. By doing so, the authors increased the reliability and validity of the study’s findings.

### 1.3. Nanoparticles: Classification and Characterization

Nanotechnology is extremely important in medicine in developing systems whose shape and size can be controlled to improve and individualize their physicochemical and pharmacological properties, and the systems obtained can be used for various purposes for numerous diseases, including cancer [18,19,20]. 

Nanoparticles are tiny materials having size ranges from 1 to 100 nm that can be obtained naturally or synthesized artificially, and research has shown that they are more effective than chemotherapy or radiotherapy used independently, producing minor adverse reactions [14,21,22,23]. Nanomaterials are used as biological markers, contrast agents for imaging, medical care products, pharmaceutical products, and drug delivery systems, as well as in the detection, diagnosis, and treatment of various types of diseases, including cancer [18,24,25,26].

The use of nano systems in medicine has numerous advantages, which are presented in Figure 1. In the case of oncological pathologies, the main advantage of using nanoparticles is that they can be directed to the sites of tumors by conjugating them with different monoclonal antibodies or peptide ligands that exhibit specificity for the receptors of the cells that these systems are intended to reach [27,28].

Nanoparticles can be classified based on several characteristics: size, charge, chemical properties, morphology, state(s) [32]. The main nano systems used in biomedicine are presented in Table 1.

Among the nanoparticles mentioned in Table 1, in the following, we will present dendrimers in more detail as nanoparticles with multiple applications in the oncological field. 

## 2. Dendrimers: Polymer-Based Nanoparticles

Our planet is full of various structures that have a dendritic, branched architecture, from neuronal dendritic branches to the rich branching of tree roots. Some researchers believe that these dendritic structures have evolved over time due to the need for highly complex and efficient surfaces that allow for the processes of absorption, extraction, or distribution of different substances in the living world [50]. Inspired by his pastime as a horticulturist, Donald Tomalia, along with his colleagues, discovered hyper-branched molecules called dendrimers in the early 1980s [50,51]. The first dendrimers obtained were polyamidoamines (PAMAM) and are called starburst dendrimers. They are composed of a core of ethylenediamine or ammonia, which is surrounded by amino groups on the outside [51,52,53]. Unlike linear polymers, dendrimers are highly branched molecules that can be synthesized at the desired size and molecular weight, and whose monomeric units have the ability to self-organize [14,51,54,55]. Some of the existing types of dendrimers are presented throughout the following Table 2.

As previously noted, the structure of dendrimers is represented by a central core, surrounded by branches, and the outermost layer of their structure is represented by a multivalent surface. The synthesis of these molecules can be done divergently or convergently [14,57,58,59]. The two synthesis methods differ in the growth direction of the dendrimer. In the case of divergent synthesis, the synthesis begins with the formation of the central molecule, the core of the dendrimer, and then this base molecule interacts with monomers, causing the structure to grow outward [60,61]. The addition of monomers to the outside of the molecule occurs for several generations in a row, adding one layer of the first-generation dendrimer. Each layer of monomers represents a generation [51,57,61,62]. Dendrimers with few layers, from generations 0, 1, or 2, have open, asymmetric structures, and as the branches become larger (dendrimers of generation 4 or higher), the dendrimers become compactly packed, taking on a globular shape. However, the synthesis of these polymer molecules cannot be infinite due to the lack of space [51,63,64,65,66]. A problem that may arise in the synthesis of dendrimers by the divergent method is the appearance of incomplete terminal groups, and this problem can be avoided by synthesis through the convergent method, in which the dendrimer is synthesized starting from the final branches. When a structure of the desired size, formed from branched monomers, is reached, it attaches to a core molecule. This method does not carry the risk of defects appearing in the final structure of the dendrimer, and purification is easy to achieve, but it cannot be used for the synthesis of high-generation dendrimers [51,67,68]. After the synthesis of dendrimers, both in their core and between the branches formed by the monomers, cavities and channels will form. The interiors of dendrimers, as well as the groups on the surfaces of these structures, can be loaded with various medicinal substances [14,51,62,69,70,71,72]. In addition, the free branches of dendrimers can be linked to other molecules in the class of nanomaterials, radioligands, or other functional molecules that reduce cytotoxicity and increase the biocompatibility of the polymer in the body. In this way, by attaching different ligand molecules, dendrimers can be targeted to different tissues [50,73]. Due to the numerous advantages of dendrimers, these polymeric molecules are increasingly being used in medicine, providing utility in the diagnosis and treatment of various conditions, directing drug substances in the body, or increasing the efficacy of other therapies [74,75,76,77,78,79,80].

Some of the strengths of the use of dendrimers have already been noted, and Table 3 summarizes the advantages of these polymer-based nanoparticles and the advantages of using dendrimers in cancer treatment.

Although dendrimers have multiple advantages for use in biomedicine, these structures also present some disadvantages. One of these is the toxicity that dendrimers have in biological systems. Due to the terminal NH2 groups and the cationic charge on the dendrimer surface, they can interact with the negatively charged cell membranes, producing cell lysis and implicitly cytotoxicity [82,83]. It seems that PAMAM dendrimers are among the most toxic, and their toxicity depends on the dendrimer generation, the lower-generation ones being less toxic. PAMAM dendrimers whose surface is modified with anionic molecules such as carboxyl groups or with PEG polymers are less or not at all toxic. It was also proven that the modification of the surface of the dendrimer so that the amino groups are replaced with aldehyde groups led to lower toxicity. Similar to PAMAM dendrimers, PPI dendrimers also show toxicity [82,83]. 

The toxicity of dendrimers is generally characterized, in addition to cytotoxicity, by hemolytic toxicity and hematological toxicity. The hemolytic toxicity is caused by the interaction of the free cationic terminal groups of the dendrimers with the red blood cells (RBC), an interaction that determines hemolysis [82]. For instance, Bhadra et al. developed fourth-generation PAMAM dendrimers for the release of an anticancer drug and found that the hemolytic toxicity of these dendrimers was around 15.3–17.3% [82,84]. Subsequently, Asthana et al. had similar results when they evaluated the same dendrimers for hemolytic toxicity and observed up to 18% toxicity [82,85]. Analyzing different generations of PPI dendrimers, a value of 35.7% hemolysis was observed for the fourth-generation dendrimers and 49.2% hemolysis for the fifth-generation dendrimers [82].

Hemolysis caused by cationic dendrimers directly influences the hematological parameters—in this case, it is a hematological toxicity. Analyzing the effect of the dendrimer PPI on blood parameters such as the number of white blood cells, the number of red blood cells, the concentration of hemoglobin, the hematocrit, and the mean corpuscular hemoglobin, a significant decrease in the number of red cells and a significant increase in the number of white blood cells were observed. Moreover, the concentration of hemoglobin, the average corpuscular hemoglobin, and the hematocrit decreased dramatically according to Agashe et al. [82,86]. 

Dendrimer toxicity in vivo is rarely studied, but Roberts et al. studied the in vivo toxicity of third-, fifth-, and seventh-generation PAMAM dendrimers in Swiss Webster mice and observed that only seventh-generation PAMAM dendrimers produced biological complications. The authors concluded that dendrimers do not present properties that prevent their use in biological applications. However, in order to avoid any of the possible adverse effects, it is necessary to modify the surfaces of the dendrimers so that they are more biocompatible [82,87].

Besides the disadvantage of the toxicity of dendrimers, the possibility of these structures being immunogenic was also investigated, but it seems that immunogenicity was not identified—it was very weak. Agashe et al. investigated the immunogenicity of 5.0 G PPI dendrimers in Balb/C mice using ELISA to monitor the antibody titer and concluded that the dendrimers did not elicit any detectable humoral immune response under the experimental conditions. Thus, dendrimers are treated by the host’s immune system as “native” and this is an advantage for their use in drug delivery [82,86].

Due to their properties of encapsulating and transporting different molecules, but also the possibility of modifying their surface so as to present a high degree of biocompatibility, dendrimers can be used in cancer diagnosis.

## 3. The Use of Dendrimers in Cancer Diagnosis

Cancer encompasses a group of invasive diseases, which is why the early diagnosis of these diseases is extremely important in order to initiate effective treatment as early as possible. Cancer diagnosis can be established through many methods, two of which are magnetic resonance imaging (MRI) and computed tomography (CT) [88,89,90,91]. MRI is a technique used to obtain anatomical images of the internal organs and the vascular tree [51,92,93]. The use of contrast agents to obtain these images significantly improves their quality, and the administration of contrast agents can be done using dendrimers [1,56,88,94]. Contrast agents commonly used in MRI are paramagnetic metal cations of gadolinium, such as Gd(III)-N,N′,N″,N‴-tetracarboxymethyl-1,4,7,10-tetraazacyclododecane (Gd(III)-DOTA) and Gd(III)-diethylenetriamine pentaacetic acid (Gd(III)-DTPA), or other derivatives thereof. However, due to their low molecular weight, these agents can diffuse from blood vessels and have a short circulation time in the body, and differentiated targeting towards diseased and normal tissues is inefficient. This disadvantage can be minimized by attaching gadolinium cations to the surfaces of dendrimers [51,56,63,70,88,95,96].

The efficiency of using dendrimers conjugated with contrast agents has been demonstrated by Bourne et al., who tested the efficiency of gadolinium-conjugated dendrimers in imaging the pelvic blood vessels of rabbits. They obtained much clearer images and much better contrast between blood vessels and the soft tissues around them when using gadolinium dendrimers [51,97]. Furthermore, PAMAM dendrimers labelled with Gd have been used to evaluate the development of lymphatic vessels in tumors [88,98,99].

Other authors have noted that these dendrimer systems could be used as molecular probes to amplify the signals of contrast agents when they reach a tumor microenvironment. This is achievable due to the ability to modify the surface properties and compositions of dendrimers by attaching specific antibodies or ligands targeting tumor receptors [95,100,101,102]. Dendrimers can also be covered with a transducer film functionalized with receptors for certain biological analytes. When the analytes bind to specific receptors, the transducer film will be mechanically stimulated and will produce a signal. In this way, cancer biomarkers can be detected [103,104,105,106].

PEG-covered PAMAM dendrimers have been used for subsequent conjugation with antibodies, folic acid, or biotin molecules, serving for specific capture of circulating tumor cells [103,107,108]. Additionally, the surface of the dendrimer can be covered with DNA/RNA or biotinylated antibodies to detect cancer antigens [103,109,110].

Thus, dendrimers can be used in many ways for the diagnosis of cancer, with the possibility of combining multiple utilities of these macromolecules for a single investigation, such as dendrimers used both in targeting the tumor and improving the images obtained by MRI.

In addition to the uses mentioned so far in cancer diagnosis, dendrimers can also be used for immunodiagnosis. Immunodiagnosis is based on the generation of signals that can be easily visualized when there is an antigen–antibody interaction between certain target molecules. The use of antibodies in immunodiagnostics allows the attachment of a single group that emits a fluorescent signal when the antigen–antibody complex has been formed, but, using dendrimers, a large number of signal molecules can be attached (for example, fluorescein) and the fluorescence signals are greatly improved; thus, immunodiagnosis is influenced by the density of molecules capable of emitting light signals [111].

A study analyzed the fluorescent signals emitted by antibody–fluorescein conjugates compared to the fluorescent signals emitted by antibody–dendrimer–fluorescein complexes and it was proven that the intensity and clarity of the fluorescence signals was significantly stronger when dendrimers were used [112].

The aim of another study was to synthesize a complex immunodiagnostic device based on dendrimers; PAMAM dendrimers with a ferrocene core were used, i.e., dendrimers that were coated with cysteamine-modified gold electrorides. The molecules obtained had the function of providing the analytical redox signal generated by the ferrocene fragments and immobilizing the prostate-specific antibody (PSA) with the help of the primary amino group on the surface of the dendrimer. The best results were obtained using first-generation dendrimers, with these complexes detecting PSA from 10 pg/mL to 100 ng/mL [113].

As mentioned previously, one of the greatest advances in the field of cancer is the use of molecules that can precisely target the tumor microenvironment, and dendrimers, in addition to their use in diagnosis, can also be used in the targeting and treatment of tumors.

## 4. Targeting and Treatment

One advantage of dendrimers is their ability to be synthesized with specific characteristics depending on the intended application. This is especially important for obtaining dendrimers that can be distributed to the sites of tumors and can transport anticancer drugs for treatment. Dendrimers can target anticancer drugs through encapsulation, covalent bonding, or electrostatic interactions, and drugs can be stored either inside the dendrimers or on their surface functional groups [114,115,116].

In the case of cancer, the targeted delivery of dendrimers is crucial to reduce side effects on healthy tissues such as internal organs and bone marrow, which can occur when chemotherapy drugs are administered freely [56,88]. Dendrimers can be conjugated with both drugs and targeting molecules, such as monoclonal antibodies, folic acid, or various peptides, to achieve more specific targeting [56,103,117,118,119]. Additionally, poli (ether hydroxylamine) dendrimers have been used to improve the water solubility of poorly soluble anticancer drugs [76,103,120]. 

Dendrimers can be distributed to specific targets via passive or active pathways [14,56,88,121,122,123]. The passive pathway is based on the accumulation of PEGylated dendrimers in tumor tissues due to the permeability and retention effects of tumors [124]. Tumors have irregular vascularization formed through tumor angiogenesis, and lymphatic drainage is inefficient, which leads to the retention of dendrimer macromolecules in the tumor microenvironment [14,27,56,88,125,126,127]. Active targeting is achieved by conjugating dendrimers loaded with drugs with different specific targeting molecules, which mediate interactions with specific cell receptors [14,56,88,123,126].

Dendrimers are used to transport a variety of pharmaceutical substances through encapsulation [128]. PAMAM dendrimers in which cisplatin, an anticancer drug, was encapsulated showed slower release of the drug compared to its free administration. Additionally, it was observed that they accumulated in solid tumors and produced lower toxicity [70,129,130]. The slower release of encapsulated substances was also observed in the case of PAMAM dendrimers with silver [70,131,132].

A study that examined the encapsulation behaviour of adriamycin and methotrexate in third- and fourth-generation dendrimers conjugated with monomethyl ether poly(ethylene glycol) chains showed that the most efficient encapsulation (6.5 molecules of adriamycin and 26 molecules of methotrexate per dendrimer) occurred in G4-PAMAM dendrimers with PEG2000 chains. It was also observed that the drug was released slowly when the medium had low ionic strength but rapidly in an isotonic medium [70,133].

The conjugation of nanoparticles with polyethylene glycol (PEG) is important to prolong their circulation time and prevent their destruction by the host’s immune system [134,135,136,137]. When the anticancer drug 5-fluorouracil was encapsulated in PEGylated dendrimers, namely G4-PAMAM conjugated to carboxymethyl PEG5000, it was released more slowly and had lower hemolytic toxicity compared to encapsulation in dendrimers without PEG on their surfaces [84].

The conjugation of dendrimers with folic acid is a good targeting method for tumors, as numerous tumor cells overexpress folic acid receptors [138]. The rapid divisions that occur in cancer cells require increased amounts of folic acid, which is a source of carbon necessary for DNA synthesis. Therefore, as many types of cancer overexpress folic acid receptors, the folic acid molecules on the surfaces of dendrimers can bind to these receptors and be internalized into the cell [56,139,140]. G5-PAMAM dendrimers conjugated with folic acid, whose free amine groups were covered with glycidol, which subsequently reacted with methotrexate, resulted in the much slower release of the drug compared to the release produced by the free administration of the drug. These dendrimers showed high specificity for human epithelial carcinoma cells [88,141,142].

Dendrimers that encapsulate certain peptides can be used as cancer treatment vaccines. The administration of dendrimers conjugated with Epitope Pan DR-binding (PADRE) peptides and ovalbumin (OVA) plasmids as a vaccine inhibited tumor growth. Double vaccination with this complex in C57BL mice resulted in the inhibition of tumor growth in 100% of cases, while animals that received plasmids encoding OVA by electroporation only showed a slowdown in tumor growth in 60% of cases [103,143,144].

The use of monoclonal antibodies in dendrimer synthesis has the role of specifically targeting tumor cells that express certain antigens [145,146]. For example, G5-PAMAM dendrimers on the surfaces of which a specific antibody to PSMA antigens—which are overexpressed in prostate cancer—was inserted specifically bound to PSMA-positive cells, and the conjugate was internalized into these cells [88,147]. 

Another category of molecules that can be conjugated with dendrimers to target tumor cells is aptamers. Aptamers are single-stranded oligonucleotides known for their increased ability to bind to target cells. They are known for being resistant to a wide range of temperatures and to different pH values, and they are extremely stable and non-immunogenic and have a low production cost, which is why they are preferred instead of antibodies. Although they are not considered by the body as a foreign structure, aptamers cannot undergo endonuclease degradation, which is why the conjugation of aptamers with dendrimers is an effective therapeutic method [148,149]. 

We previously predicted that dendrimers are structures that cause toxicity, and coating them with aptamers reduces dendrimer toxicity. Practically speaking, the aptamer–dendrimer complex is an advantage for each of these two separate structures [148].

Taghdisi et al. analyzed the therapeutic action of dendrimers based on aptamers with a double targeting strategy. The dendrimers were conjugated with MUC1 and AS1411 aptamers, but also encapsulated epirubicin, an anticancer drug. The MUC1 aptamer selectively binds to transmembrane glycosylated mucin-1 (glycoprotein), which is overexpressed in many tumors, while the single-chain AS1411 aptamer specifically binds to the nuclear membrane protein nucleolin, which is overexpressed on the plasma membranes of tumor cells. The obtained dendrimer–aptamer conjugate delivered epirubicin to MCF-7 breast cancer cells and C26 colon carcinoma cells. Using flow cytometry, it was shown that the dendrimer–aptamer complex was specifically internalized into tumor cells (MCF-7 and C26) by receptor-mediated endocytosis using the MUC1 aptamer and by using the non-standard mechanism of the AS1411 aptamer, leading to the accumulation of the complex in the nucleus. No internalization was observed in Chinese Hamster Ovary (CHO) cells, for which the aptamers had no specificity. In addition, the internalization of the complex was better using both aptamers than in the case of using each aptamer separately. Furthermore, the antitumor action was better when the dendrimer–aptamer complex was administered than when epirubicin was administered alone [150,151]. 

As previously mentioned, in the case of oncological pathologies, targeted administration of anticancer molecules is extremely important [13,152]. In addition to precisely targeting anticancer substances, their slow release, and their long-term retention in tumors, dendrimers can be involved in cancer treatment through other mechanisms, such as the use of polyphenol dendrimers with antioxidant action. These can be targeted towards tumors to reduce oxidative stress, which is involved in the apoptotic pathway [103,153,154].

Another way in which dendrimers can be used in cancer treatment is based on neutron capture [155]. Some dendrimers (G6-G8 PAMAM, G5 PPI) have been conjugated with DTPA or DOTA chelating agents, as well as with neutron capture elements based on gadolinium, and it has been observed that the administration of these complexes to laboratory animals carrying tumors resulted in efficient anticancer activity [103,156,157]. Neutron capture agents interact with thermal neutrons and cause nucleus destruction and DNA strand breaks [158]. These agents have small sizes and can easily diffuse from tissues, but their use through conjugation with dendrimers results in the accumulation of larger amounts of Gd in tumors, which leads to increased anticancer effects [103,159,160].

The use of dendrimers in the treatment of cancer can also consider the direction to tumors of some molecules that absorb light radiation and whose action is toxic for tumor cells. Such use of dendrimers applies in the case of photothermal or photodynamic therapy.

## 5. Photothermal Therapy

It is known that a high body temperature is dangerous because it causes protein denaturation and cellular damage [161]. This phenomenon also applies to cancer cells, as many studies have shown that photothermal therapy based on hyperthermia can be used in anticancer therapies [77,162,163,164]. The tumor microenvironment is characterized by low oxygen pressure and a low pH due to inefficient circulation at this level. Maintaining high temperatures between 41 and 48 °C in these areas is cytotoxic to cells and will cause cell death [163,165,166]. Due to inefficient circulation, anticancer agents reach the tumor in small amounts, and, in this case, hyperthermia can be used concomitantly with chemotherapy or radiation therapy to achieve better treatment results [77,162,163,167,168]. 

Photothermal therapy involves the destruction of cancer cells by increasing the temperature of the tumor tissue following exposure to infrared light. In these cancer treatment methods, nano systems play an important role as they can be used to absorb infrared rays, thus increasing the efficiency of heat production in tumors [14,169,170,171,172].

Nano systems can be released into the tumor environment either by direct injection or by targeted delivery, and the molecules that reach the tumor environment will be used as photo-absorbents that will produce thermal energy when illuminated with near-infrared light [33,77,173,174].

Photothermal therapies are based on the use of a laser, whose power, exposure time, and wavelength may vary depending on the properties of the tumor tissue on which it will act [33,175,176]. In the case of this anticancer therapy, dendrimers can be used as devices to absorb light radiation [103,177]. Li et al. used PEG-PAMAM dendrimers, which they hybridized for the absorption of infrared rays with a gold nanorod (GNR). Single irradiation as well as administration of the dendrimers mentioned above without irradiation did not cause damage to HeLA cells derived from human cervical cancer. However, when these dendrimers with a GNR core were subjected to irradiation, the generated heat caused an increase in temperature in the irradiated tumor tissue with an NIR laser and a decrease in tumor volume [178]. Due to the frequent use of dendrimers to transport medicinal substances, they can be used for combined therapy to deliver both chemotherapeutic and photothermal therapy molecules to tumor cells [178].

Another complex study, conducted by Grześkowiak et al., showed the effects of G3-PAMAM dendrimers conjugated with polydopamine (PDA) molecules. Polydopamine has high efficiency in the process of transforming near-infrared light into thermal energy [179]. The study results showed that coating PDA spheres with PAMAM dendrimers reduced the viability of cancer cells compared to the administration of pure PDA spheres. The WST-1 test and the staining of live/dead cells showed that cancer cells still had a high survival rate following the administration of G3-PAMAM-PDA, but their viability was significantly reduced after laser irradiation. After 48 h, complete cell death was observed in irradiated cancer cells containing dendrimers in a concentration of at least 10 µg/mL [180]. The main mechanism used in photothermal therapy is presented in Figure 2.

## 6. Photodynamic Therapy

Photodynamic therapy (PDT) is a therapeutic strategy based on the production of reactive oxygen species, which subsequently cause cell death [181]. For this mechanism, a photosensitizer, oxygen molecules, and light are necessary. Essentially, a photosensitizer must reach the level of the tumor tissue, and then the area is stimulated with an appropriate wavelength, and the photosensitizer will form singlet oxygen species. Intracellularly, reactive oxygen species will accumulate, and increased oxidative stress will determine cell death through apoptosis or necrosis [33,56,181,182,183,184,185,186]. The principle of photodynamic therapy is presented in Figure 3.

PDT can be used to destroy tumor cells, for tumor neovascularization, or to increase the inflammatory response and attract immune cells to the tumor environment [56,187,188,189]. Since it is difficult to obtain deep light penetration, this therapy is greatly improved by using targeted delivery systems. Due to retention processes, the molecular weight, and hydrophilicity, dendrimers used to deliver photosensitizers have yielded promising results for photodynamic therapy [56,190,191].

One photosensitizer agent is phthalocyanine. PAMAM dendrimers that have encapsulated phthalocyanine in the central core have produced cell death to a high percentage following irradiation with halogen light for 10 min. The administration of PAMAM dendrimers with phthalocyanine without irradiation did not affect the viability of cancer cells [103,192].

Photodynamic therapy is non-invasive, does not produce lesions or scars at the site of application, and can be used multiple times, but it is not effective for deeply developed cancers that have metastasized in multiple areas of the body [33,103,181,193].

Kojima et al. compared PEG-PAMAM dendrimers to PEG-PPI dendrimers, both loaded with protoporphyrin IX (PpIX). Their results showed that PEG-PPI dendrimers produced greater toxicity after irradiation compared to PEG-PAMAM dendrimers. The PEG-PPI-PpIX complex was more stable and generated the release of singlet oxygen, and PpIX molecules reached the mitochondria, generating increased phototoxicity [194].

Another means of using dendrimers in anticancer therapies is represented by their conjugation with molecules of genetic material, an upcoming aspect that will be presented.

## 7. Gene Transfection

Dendrimers can be used as delivery vectors for genetic material (Figure 4). They can transfer DNA or RNA molecules for cancer treatment [195,196]. PAMAM dendrimers with amino end groups can interact with the phosphate groups in nucleic acid molecules, forming complexes that will be directed to tumors and allow the transfer of genetic material through endocytosis into cancer cells and then into their nuclei [51,56,101,197,198,199].

A transfection system based on dendrimers is commercially available under the name SuperFectTM. These dendrimers are stable and can transport a larger amount of genetic material than viral vectors, and the release into the nucleus has been shown to be more efficient than when liposomes are used [51,200,201].

It seems that for gene transfection, dendrimers with an excess of amino groups compared to the phosphate groups of the genetic material are more efficient [56,202].

A therapeutic strategy to stop the progression of malignant tumors can target angiogenesis [203,204,205,206,207]. For this therapeutic strategy, dendrimers associated with anionic oligomers were used to release angiotensin and genes that determine the production of tissue inhibitor of metalloproteinase—TIMP-2. Gene transfer to breast cancer tissue significantly reduced the proliferation of endothelial cells [204,208,209].

PAMAM G1-G4 Tomalia-type dendrimers with a di-n-dodecylamine molecule as the core were synthesized. They formed complexes with DNA, and dendrimers from G2-G4 were able to cross cell membranes and deliver DNA efficiently [56,210,211].

In addition to PAMAM dendrimers, PPI dendrimers have also been used in gene transfection. G1-G5-PPI dendrimers conjugated with PEG molecules and containing cationic ammonium interior groups were used to transfect ss-DNAzyme oligomers into ovarian carcinoma cells. The use of this complex resulted in efficient and stable gene delivery [103,212].

As noted earlier, dendrimers are suitable for combining different therapeutic strategies. When dendrimers are synthesized to contain different genetic material fragments, their interiors can also include pharmaceutical substances such as doxorubicin. In this way, dendrimers will target tumors and allow for both chemotherapy administration and the delivery of genetic material to suppress tumor function [1,103,207,213].

The mechanism of gene therapy using dendrimers is presented in Figure 5.

## 8. Conclusions

Dendrimers are nanoscale drug delivery systems for anticancer drugs that can target the tumor site. These biocompatible nano systems have properties that can be used for diagnostic purposes, transdermal drug delivery, and medication conveyance in cancer. 

The majority of current anticancer drugs do not differentiate cancerous cells from normal cells, and they lead to systemic toxicity and side effects. Dendrimers can be successfully used for gene therapy or for delivering the antineoplastic agent in cancerous cells, by active targeting, without causing any toxicity. Dendrimers can be directed specifically toward cancer cells (e.g., by antibodies specific for tumor-associated antigens) and the ingestion into the cell can be receptor-mediated. Therefore, they will ensure selective intratumoral accumulation and reduced systemic toxicity.

Including dyes or other materials (genetic materials, targeting agents) within the dendrimer structure (by encapsulation, complexation, or conjugation) can make dendrimers useful as diagnostic tools or for tumor localization and therapy monitoring.

Dendrimers have the potential to be used as theragnostic particles, with both diagnostic and therapeutic functions at the same time (drug delivery and therapy monitoring for optimum drug dosage and tumor growth).

## 9. Future Perspectives

Dendrimers’ approach of DNA-based nanomaterial delivery in cancer therapy could significantly improve cancer diagnosis and therapy.

Dendrimers can represent part of a multimodal nanoparticle that could ensure the precise delivery of antitumor drugs and could double the efficiency of diagnosis and therapy.

Manipulating the architecture of the dendrimers, their properties (efficiency of delivery and biocompatibility) can be increased to a more efficient level, ensuring the enhancement of the bioavailability for problematic drugs.

Continuous research in nano-oncology can lead dendrimers to become the newest class of curative anticancer therapeutic agents.

## Figures and Tables

**Figure 1 pharmaceutics-15-01406-f001:**
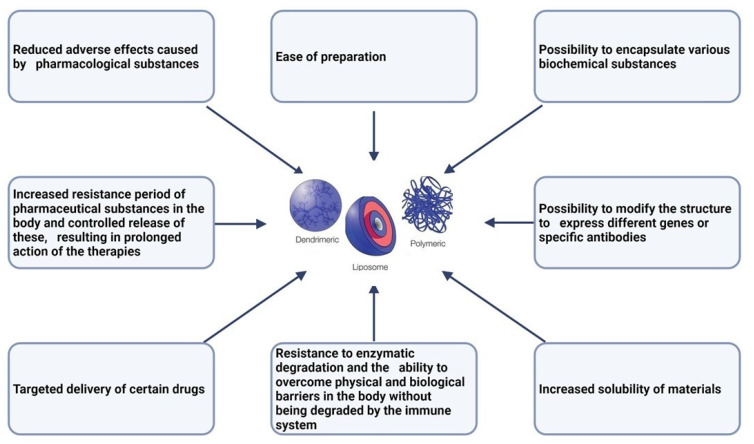
Advantages of nano systems in medicine [18,29,30,31].

**Figure 2 pharmaceutics-15-01406-f002:**
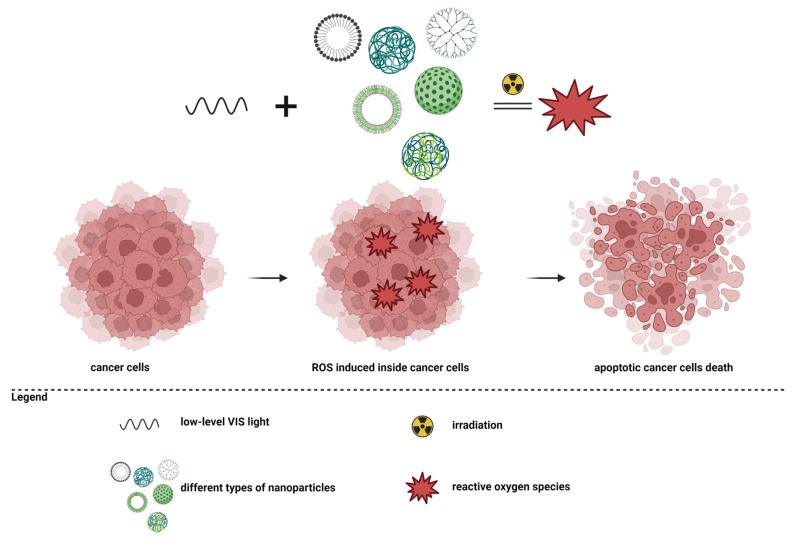
Photothermal therapy in cancer.

**Figure 3 pharmaceutics-15-01406-f003:**
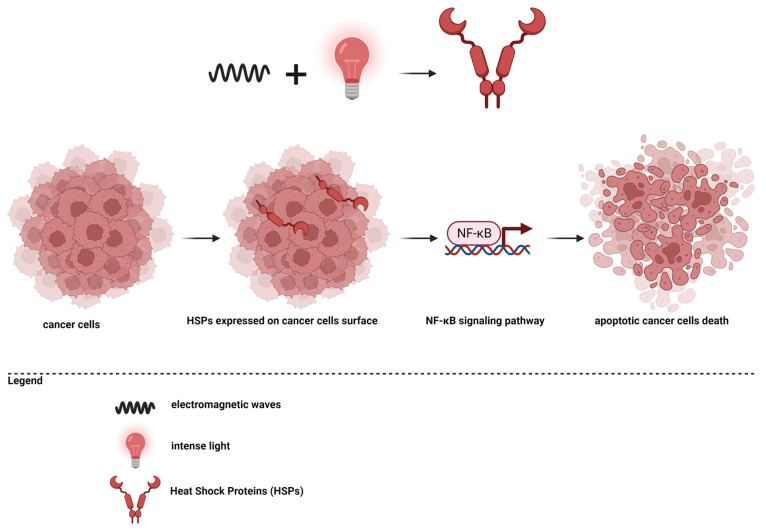
Photodynamic therapy in cancer.

**Figure 4 pharmaceutics-15-01406-f004:**
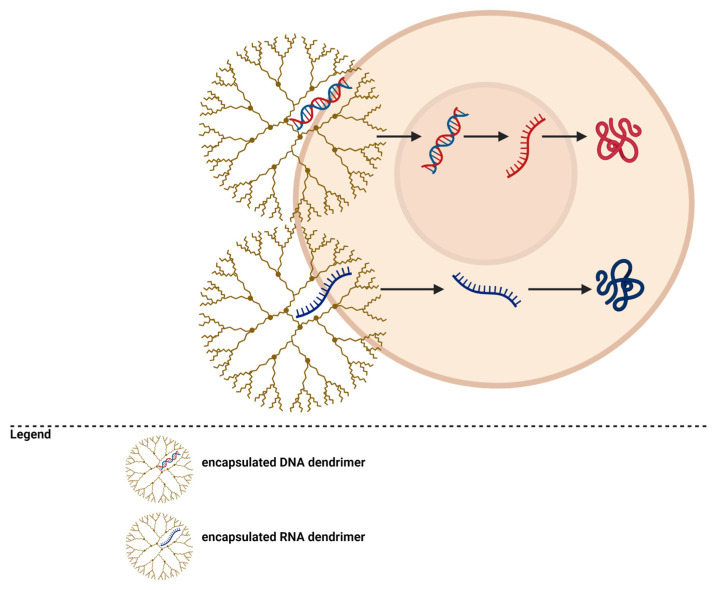
Dendrimers for gene transfection.

**Figure 5 pharmaceutics-15-01406-f005:**
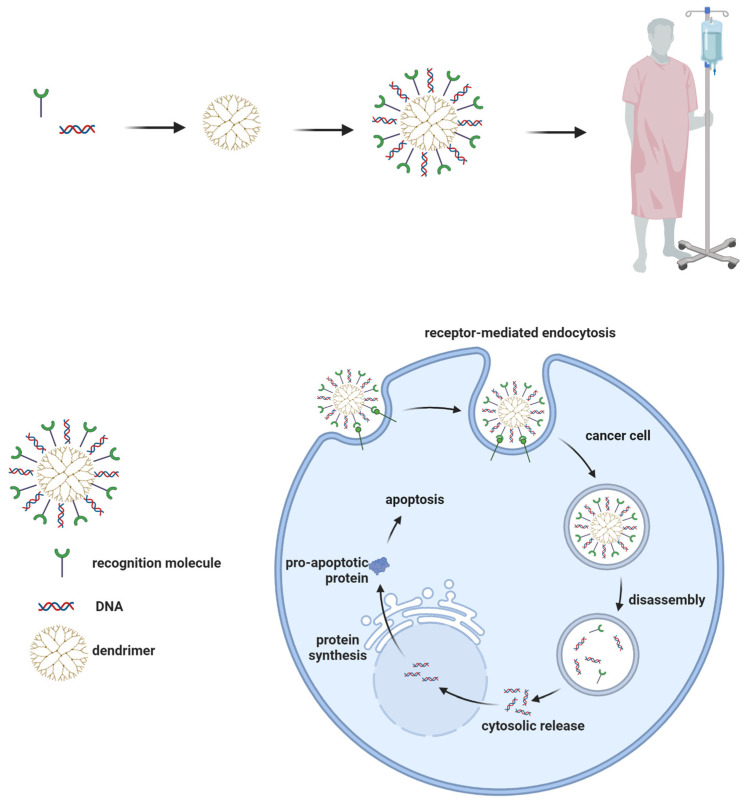
Dendrimers’ mechanism for gene therapy.

**Table 1 pharmaceutics-15-01406-t001:** Nanoparticles in drug delivery: a comparative analysis of polymeric nanoparticles, carbon nanotubes, liposomes, gold nanoparticles, micelles, solid lipid nanoparticles, and dendrimers.

Type of Particle	Characteristics and Properties	Strengths	Weaknesses
Polymeric nanoparticles	- efficient for targeted drug distribution - possibility to modify shape, composition, size, and surface characteristics	- stability - high therapeutic efficiency - large encapsulation capacity	- toxic degradation - difficult encapsulation for hydrophilic drugs [33,34,35,36,37]
Carbon nanotubes	- cylindrical shape with a closed end, suitable for antimicrobial agents and gene delivery - electrical conductivity	- resistance - high specific surface area	- insoluble in aqueous and organic solvents [14,33,36,38]
Liposomes	- externally formed from a double hydrophobic phospholipid bilayer that can be modified to reduce side effects, while the interior is represented by an aqueous core in which drugs, proteins, genes, or peptides can be encapsulated - specific drug transport without degradation	- high biocompatibility - high permeability of drug distribution	- sensitivity to extrinsic and intrinsic stimuli [14,33,36,39,40,41,42]
Gold nanoparticles	- exhibit unique optical properties and have the ability to conjugate with antibodies - water solubility - high biocompatibility	- extremely stable - low toxicity	- biological stability may decrease in vivo [33,36,43,44,45]
Micelles	- colloidal particles with a spherical shape, with a polar exterior surface and a non-polar interior - prolonged drug release	- high biostability - high drug loading capacity	- only used for lipophilic substances - low stability in blood [14,36,46,47]
Solid lipid nanoparticles	- formed from a monolayer of phospholipids on the outside, which line a hydrophobic core	- can surpass the blood–brain barrier - no problems with respect to large-scale production and Sterilization	- unpredictable gelation tendency - unexpected dynamics of polymeric transitions - sometimes burst release [33,48,49]
Dendrimers	- highly branched polymeric molecules that have self-organizing capacity and are used in imaging, cancer therapies, and gene therapies - controlled synthesis	- high biocompatibility	- cytotoxicity [14,33,36]

**Table 2 pharmaceutics-15-01406-t002:** Types of dendrimers [2,56].

Type of Dendrimer	Characteristics
Chiral dendrimers	- built on different conditional branches and having a chiral core in the center - these dendrimers are optically active
PAMAM dendrimers	- dendrimers are most frequently used in the distribution of medicines - the surface is modified to become non-immunogenic and it has a high capacity to encapsulate different compounds, due to the amide bonds and the compositionrich intertiary amines
PPI dendrimers	- formed from poly-propylene imine - used in the delivery of medicines
Tacto dendrimers	- consists of a central dendrimer that is conjugated with several dendrimers of different types at the periphery - perform functions of complex nanodevices
Hybrid dendrimers	- consisting of several dendritic combinations and linear polymers, in different forms
Peptide dendrimers	- they are dendrimers that present peptides on the surface or dendrimers that include amino acids - they have an important role in various medical therapies, including in the delivery of medicines
Glicodendrimers	- dendrimers that incorporate carbohydrates either in their core or on the surface, or are built entirely from carbohydrates - they are used for cell recognition studies, the targeting of contrast agents for MRI, and the delivery of drugs and genes

**Table 3 pharmaceutics-15-01406-t003:** Advantages of dendrimers.

Advantages of Dendrimers	Advantages of Using Dendrimers in Cancer Treatment
Ability to synthesize molecules with desired characteristics based on the intended purpose	
High endocytosis capacity	
High capacity for drug encapsulation	
Can be administered orally, intravenously, or in combination	
Biocompatible and biodegradable	
Ability to improve the solubility of hydrophobic drugs	
Delivery and controlled release of drugs	
Possibility of conjugation with different molecules that reduce toxicity	Possibility of attaching specific ligands to target tumor tissue and reducing cytotoxicity towards healthy cells
Globular structure→ small hydrodynamic volume	
Monodisperse architecture	
Multivalent surface	Possibility of covalent conjugation with several different anticancer molecules
	Possibility of monitoring the effectiveness of the treatment
	Treatment administration avoiding the possibility of developing drug resistance
[14,34,51,55,57,58,70,74,75]	[76,77,78,79,80,81]

## Data Availability

The data presented in this study are available on reasonable request from the corresponding author.

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
