# Peer review of "Dendrimers: Advancements and Potential Applications in Cancer Diagnosis and Treatment—An Overview"

_pharmaceutics, 2023, doi:10.3390/pharmaceutics15051406_

Round 1

Reviewer 1 Report

The manuscript entitled summarized the recent advances of the application of dendrimers in cancer diagnosis and various treatment. The review showed certain importance. However, there are some issues need to be addressed before publishing in Pharmaceutics.

1.     In the Introduction section, the authors emphasized the importance of nanotechnology for tumor targeting. However, there was no further explanation of the targeting mechanism in the following section. The authors are encouraged to add this information in the revised version.

2.     The authors wanted to show the advantages of nanosystems in medicine in Figure 1. However, the authors only put dendrimer in the figure as a representative of nanoparticles. It is better to include other nanomaterials in the figure to avoid misunderstanding that the nanosystems are only dendrimers.

3.     In Table 1, the weakness of Solid lipid nanoparticles was missing. There was only references in this column. Please check and confirm it.

4.     In Figure 2, the authors indicated that low cost of synthesis is one of the advantages of dendrimers. But we all know that the synthesis of dendrimers is more challenging than traditional dendrimers. What does the “low cost of synthesis” mean?

5.     The author list the advantages of dendrimers and using dendrimers in cacner therapy in Figure 2 and 3, respectively. The presentation of these two figures looks like a table, rather than a figure. And why the authors add different background color to the text in the figure? Different colors represent for different classification of the advantages?

6.     Many figures (Figure 5-7) were not mentioned in the text. It is better to have a check and complete them in the text.

7.     The transitions among sections in the text are abrupt and difficult to understand. The authors need to improve it.

Reviewer 2 Report

This review will highlight the advantages of dendrimers in treating oncological conditions and the multiple uses for which these structures can be used. The manuscript needs major and minor corrections.
1. The English text of Manuscript needs corrections.
2. The purpose of writing this manuscript is not written in the introduction section.
3. The introduction is very short and no explanation about dendrimers is given in it.
4. In line 59, it is written that "Nanoparticles are biochemical structure" This sentence is not correct and they do not have a biochemical structure more than nanoparticles.
5. The legend of Figure 1 should be corrected, as well as in the corresponding text because the figure shows dendrimers.
6. Figure 2 and 1 should be combined and converted into a table.
7. Subtitle 4 should be written in full, for example, "the use of dendrimers in cancer diagnosis"
8. Subtitle 4 should be written in full
9. Figures 3, 5, 6 and 7 are not mentioned in the text.
10. A separate section should be considered for the toxicity and disadvantages of dendrimers in the clinical treatment of cancer.
11. A separate section should be considered for future perspective.
12. Consider a small section for different types of dendrimers.
13. There are various references that writers can use to make the manuscript more productive, for example:
PMID: 34757195
PMID: 23678447
PMID: 25555748
14. The search strategy is not written.

15. It is recommended to write about multifunctional dendrimer-based nanoprobes.
16. It is recommended to write about dendrimer-based nanosensors in immunodiagnosis.
17. It is recommended to write about aptamer-engineered dendrimer for cancer therapy

Reviewer 3 Report

The article "Dendrimers: Advancements and Potential Applications in Cancer Diagosis and Treatment - An Overview" by Andreea Crintea et al. discusses the use of various types of dendrimers for the diagnosis and treatment of cancer. At the beginning of the manuscript, the authors briefly characterize dendrimers. Further, they give examples of the use of a combination of dendrimers with contrast agents for magnetic resonance imaging and computed tomography, as well as of attaching specific antibodies or ligands targeting tumor receptors to dendrimers (diagnosis). The manuscript shows the prospects for using dendrimers in photodynamic therapy, photothermal therapy, as well as delivery vectors for genetic material and for the delivery of anti-cancer drugs. The authors have done a colossal job by analyzing more than 200 literary sources, among which there are a lot of the latest works. It is important to note that dendrimers are currently at the peak of research interest, and this article reflects this well. The article is written very briefly and in a language that is accessible not only to specialists in the field of drug delivery. Thus, it can be useful to a wide range of readers. In my opinion, the article is suitable for publication in the journal "Pharmaceutics". However, there are minor inaccuracies in it, which are more of a technical nature, which need to be corrected.

In Table 1, in the Weaknesses column for Solid lipid nanoparticles, only literature references are presented without listing the weaknesses themselves.

It also seems redundant to me to do Figures 2 and 3 separately. It is quite possible to combine them, since they are very close in nature.

There are no references to Figures 3, 5-7 in the text of the manuscript.

In Figure 7, it is not clear where the dendrimer disappears after entering the cancer cell. The drawing needs to be corrected.

Reference 72 lacks the imprint of the article (INEOS OPEN, 2021, 4(5), 176–188, DOI: 10.32931/io2122r). The patents cited in refs 201 and 202 are also incorrectly executed.

Round 2

Reviewer 2 Report

All corrections have been made and the manuscript is acceptable.